# Relationship between Income and Mental Health during the COVID-19 Pandemic in China

**DOI:** 10.3390/ijerph19158944

**Published:** 2022-07-22

**Authors:** Mingna Li, Bo Zhou, Bingbin Hu

**Affiliations:** 1School of Literature, Changchun University, Changchun 130012, China; adbell@163.com; 2School of Economy, Jilin University, Changchun 130012, China; 3School of Northeast Asian, Jilin University, Changchun 130012, China; hu1061512673@gmail.com

**Keywords:** income, mental health, education, registered residence, happiness, COVID-19, China

## Abstract

Mental health problems represent one most pressing concerns in the world, which produce costly consequences for individuals, families and society as a whole. One of the determinants on mental health during the COVID-19 pandemic is income. To complement relevant research and provide valuable recommendations for governments and societies, this study investigates the nexus between income and mental health in China by employing 8049 observations from the 2020 China Family Panel Studies. Using ordinary least squares regression we find the significant positive relationship between income and mental health, and estimate the effect of income on mental health. Furthermore, this effect is heterogeneous depending on individuals’ education level and registered residence type. Finally, individuals’ economic status and happiness are shown to be the potential mechanism through which the effect of income on mental health operates.

## 1. Introduction

The novel coronavirus disease (COVID-19) was first detected in December 2019 [1] and spread rapidly around the world. The severity of the situation caused the World Health Organization (WHO) to declare COVID-19 a global pandemic in the second week of March 2020 [2]. From the outset, the COVID-19 pandemic had a significant negative impact on people’s mental health [3,4,5]. For example, Ferrara et al. [6] reported that the COVID-19 disease and the restrictive government measures harmed the mental health of individuals around the world, causing fear, frustration, anger and a long list of complex negative emotions. This result was also confirmed by Sánchez et al. [7,8,9,10]. McAlearney et al. [11] studied the mental health status of first responders (including police, firefighters, and emergency medical services) and their influencing factors during the pandemic. Many similar studies have been conducted and can be found in the literature [12,13,14].

Though the negative impact of the COVID-19 pandemic was tested, the relationship between income and mental health was not clear yet. There were some studies on the link between human activity and mental health during the COVID-19 pandemic. For example, Quarta et al. [15] analyzed the association of physical activity and outdoor leisure time with the mental health of college students during this pandemic. Foong et al. [16] conducted a study on the relationship between Internet use and mental health in the elderly during this pandemic. Laurinaitytė et al. [17] studied the association between health risk behaviors and mental health among Lithuanian university students during this pandemic. In addition, studies have shown that persistent loneliness and social isolation have adverse effects on the mental health of college students [18]. Investigating the determinants of mental health during the COVID-19 pandemic from different perspectives was necessary, but little attention had been paid to the effect of income on mental health.

Different from the existing research, the marginal contribution of this paper lies in the following three aspects. First, this paper investigated the effect of income on mental health in a developing country: China. We used data from the 2020 China Family Panel Study (CFPS) [19], whose stratified and multi-stage sampling design enables the sample to represent almost 95% of the Chinese population. Second, we examine the heterogeneity of effects by dividing the sample into different education levels and type of residence registration. These results help us understand the heterogeneity of the effects of personal income on mental health. Third, the existing research fails to provide mechanisms on how income affects mental health. This paper reveals that income can affect mental health through impact on economic status and happiness.

## 2. Literature Review

Income has a crucial impact on people’s living conditions. Therefore, it has been widely studied, with researchers exploring its impact on dietary nutrition and aspects such as vaccinations, renewable energy, attention deficit and hyperactivity disorder, physical activity, and household water consumption [20,21,22,23,24,25]. It is clear that income affects people’s lives and physical health; however, its impact on mental health has rarely been systematically analyzed. The level and change of income can affect people’s state of depression, anxiety, sleep, interpersonal relationships, etc. [26,27,28,29].

The importance of mental health is confirmed by the World Health Organization [30], which defines health as “the state of complete physical, mental and social well-being and not just the absence of disease or infirmity”. In addition, mental health disorders account for a growing share of the global disease burden. The two most common mental health conditions, depression and anxiety [31,32], not only cause a great deal of suffering and health loss, but also lead to a loss of economic output [33]. Therefore, exploring the factors that contribute to mental health problems is critical to the well-being of individuals, their families and society as a whole.

Given the importance of mental health, mental health issues have attracted the interest of scholars. Existing mental health research mainly focuses on its influencing factors. On the one hand, various studies have analyzed the impact of environmental pollution as well as natural disasters on mental health. For example, Zhang et al. [34] found that air pollution reduces hedonic happiness and increases the rate of depressive symptoms. Compared with air pollution, natural disasters are difficult to control and deal with for human beings. Some studies investigate the possible impact of specific natural disasters on mental health. Llorente-Marrón et al. [35] found that earthquakes have a negative impact on the physical and mental health of victims. The study by Gissurardóttir et al. [36] suggested that exposure to volcanic eruptions may lead to mental health disorders.

On the other hand, many studies focus on the impact of socio-economic factors on mental health, such as Katiki Reddy et al. [37], whose research shows that the population mental health of men has deteriorated within 2 years of the onset of the current recession. The results of Miquel et al. [38] suggest the potential adverse consequences that job and income loss, together with financial concerns, might have on workers’ mental health. Some scholars have studied the impact of income on mental health. Thomson et al. [39] found that changes in income may affect mental health; as income increases, mental health improves, particularly where individuals move out of poverty. Using data from the Health and Retirement Study, research by Golberstein et al. [40] corroborates this notion, showing that higher income positively impacts mental health by reducing psychosocial stress associated with financial hardship. A study [41] in China also found that higher income was associated with better mental health due to better life satisfaction, living conditions, and access to healthcare. Furthermore, Ao et al. [26] found that very rich people had the lowest level of anxiety during the COVID-19 pandemic. Conversely, as income decreases, mental health deteriorates. Jenkins et al. found that people with lower incomes are more likely to develop mental disorders [42]. The results of several other studies have also confirmed this view [43,44]. However, the study by Elliott et al. [45] showed that, unlike in the UK, lower household income was not associated with poorer mental health in Italy.

The above literature explores the effect of income on mental health from different perspectives, and puts forward many constructive insights, providing a lot of theoretical material for this article. However, there are inconsistent conclusions on the effect of income on mental health during the COVID-19 pandemic, and not much attention has been paid to the mechanisms of its impact. This article attempts to fill a gap in the relevant literature on the relation between income and mental health during the COVID-19 pandemic. Therefore, on the basis of previous studies, this paper explored the effect of income on Chinese mental health and its potential impact mechanism. Moreover, this paper studied the heterogeneity of the effect among groups with different education levels and different types of residence registration. Based on the above analysis, this paper proposes the following three hypotheses.

**Hypothesis** **1** **(H1).***Income has a**significant positive effect on individual’s mental health, but this effect can be non-linear*.

Income may affect individual’s life through different aspects [46]. For example, lower income brings huge economic pressure, which makes individuals face more psychological crises, showing psychological symptoms such as anxiety, loneliness, and sensitivity to human relationships. On the other hand, higher income increases people’s financial ability to afford a spending budget for the living standards and professional services that are required to maintain healthy lifestyles, improving factors such as depression, impaired relationships, and food security [26,27,28,29,47]. However, existing research shows that for low-income groups and middle- and high-income groups, the impact of income changes on their mental health may be different [39], and this group difference shows that income has a non-linear effect on mental health. Based on the above analysis, we propose Hypothesis 1.

**Hypothesis** **2** **(H2).***The impact of income on an individual’s mental health is heterogeneous and depends on an individual’s education level and type of residence registration*.

Sánchez et al. [7] argued that COVID-19 had a greater impact on the mental health of less-educated participants. Well-educated people tend to have higher incomes, healthy diets, and an optimistic outlook on life. Well-educated people have more social and economic resources than less-educated people. Those well-educated individuals can better cope with the negative effects of low incomes [47]. In addition, Wen’s [48] research shows that there are still class differences in the mental health of the elderly, and the degree of mental depression in the elderly with agricultural household and low education level is higher. The effects of absolute poverty and relative poverty on the mental health of the elderly are different between urban and rural areas [49]. The impact of income on mental health may not be the same for people with different educational levels and types of residence registration. Therefore, Hypothesis 2 arises.

**Hypothesis** **3** **(H3).***Economic status and happiness are potential mechanisms by which income affects mental health*.

In terms of academic logic, only using objective variables to explain people’s subjective feelings also has its limitations. Faced with the same income, people may have completely different subjective feelings based on economic status and happiness. Existing research suggests that the rank of personal revenue in the local area has a positive effect on mental health, as people with relatively higher incomes have more positive emotions and attitudes towards life than those with relatively lower incomes [50]. Furthermore, individuals with a higher level of happiness have more positive emotions and attitude to life than those with a lower level [51]. The existing literature indicates that income has a significant positive impact on individual happiness [52,53]. Hence, we propose Hypothesis 3.

## 3. Materials and Methods

### 3.1. Study Design

The data used in this study come from the China Family Panel Study (CFPS), a nationwide, large-scale household tracking survey project involving the economy, education, family relations, health, etc., conducted by the Institute of Social Science Survey of Peking University. Since 2010, the survey has continuously tracked information from the three dimensions of the individual, family, and community every 2 years, which more comprehensively reflects the dynamic changes of China’s society, economy, population, and other issues, and it is also the most extensive continuous survey sample at present. Its stratified multi-stage sampling design enables the sample to represent almost 95% of the Chinese population, which is conducive to further heterogeneity analysis. To examine the effect of income on mental health during the COVID-19 pandemic, this study used only the 2020 baseline survey for analysis. After excluding the samples involving variable missing values and uncertain answers in this study, a total of 8049 valid observations were obtained. Ordinary least squares method (OLS) was used for econometric analysis to investigate the effect of income on mental health.

### 3.2. Statistical Analysis

Statistical analysis was performed using the econometric software STATA version 16 (StataCorp LLC, College Station, TX, USA). We report the mean, standard deviation, minimum, and maximum values of the variables in Table 1. Given that mental health is a continuous variable, we constructed the OLS to investigate the relationship between income and mental health. In our robustness check, we used the logit and probit models to estimate the effect of income on mental health. To investigate the mechanisms, we estimate first the effect of income on individuals’ economic status and happiness, and then estimated their effect on mental health. All reported *p*-values were two-tailed. The level of statistical significance was set at *p* < 0.1.

### 3.3. Variables and Descriptive Statistics

#### 3.3.1. Outcome Variable: Mental Health

The main outcome variable in this paper is the mental health status of individuals in China. In order to evaluate individuals’ mental health more objectively and comprehensively, following existing studies [54], the mental health index is derived from the 6-item short form of the Center for Epidemiologic Studies of Depression (CES-D) in the CFPS. (CES-D questions: (1) In the past week, how many times have you felt down? (2) In the past week, how many times did you feel that you were struggling to do anything? (3) In the past week, how many times have you had poor sleep? (4) In the past week, how many times have you felt lonely? (5) In the past week, how many times have you felt sad? (6) In the past week, how many times have you felt that life could not go on? Individuals were asked to indicate the frequency of their feelings on a four-scale metric—“Almost never (less than a day)”, “Sometimes (1–2 days)”, “Often (3–4 days)”, “Most of the time (5–7 days)”. These responses are assigned a value of 4 to 1, respectively.) There are six subjective questions to evaluate mental state in the questionnaire. Basing on the equal weighted sum of the standard deviation, the final mental health index is obtained. The specific formula is as follows:(1)$${\mathrm{MentalHealth}}_{i}=\sum_{j=1}^{6}\frac{x_{\mathrm{ij}}-u_{j}}{\sigma_{j}}$$

In the formula, ${\mathrm{MentalHealth}}_{i}$ represents the mental health index of the individual, $x_{\mathrm{ij}}$ represents the individual data of each variable, $u_{j}$ and $\sigma_{j}$ represent the overall mean and standard deviation of the variable, respectively. The mental health index calculated from this is −21.165 to 4.537. The higher the mental health index, the better the individual’s mental health, and vice versa.

#### 3.3.2. Independent Variable: Income and Wage

We use two measures of income as independent variables. The first variable is the individual’s annual income (Income), which is calculated as the sum of the after-tax wage income from the main job and general job in the past 12 months. The second variable is hourly wage (Wage). It measures how much an individual earns per hour worked. In order to obtain regression results with economic implications, income and wage were processed by taking the natural logarithm.

#### 3.3.3. Mediating Variable: Economic Status and Happiness

To study the mechanism by which income affects individual mental health, we selected two variables: economic status and happiness. Economic status ranges from 1–5. 1 indicates the lowest income position locally and 5 indicates the highest income position locally. Happiness ranges from 0 to 10, with 0 meaning the least happy and 10 meaning the happiest.

#### 3.3.4. Control Variables and Descriptive Statistics

In order to address the problem of selection bias caused by omitted variables, this study selects as many control variables as possible that affect mental health. We mainly included the following control variables: sex, age, marital status, registered residence, smoke, drink, short video and online game. Furthermore, we control for medical expenses, education, identity of worker, and type of work. The descriptive statistics of the variables used in this paper are shown in Table 1. It can be seen that the average age of individuals in the sampled sample is 39.402 years old, and 59.8% of them are male. The mental health index of the sample ranges from −21.165 to 4.537. In addition, the average of annual earnings is 10.358 and the average of hourly wage earnings is 2.541.

### 3.4. Model Settings

The ordinary least squares method is used to estimate the effect of income on individual mental health. The specific formula is as follows:(2)$${\mathrm{MentalHealth}=\beta }_{0}{+\beta }_{1}\mathrm{Income}+\sum_{1}^{j}\beta_{j}{\mathrm{Control}}_{j}+\varepsilon$$
(3)$${\mathrm{MentalHealth}=\beta }_{0}{+\beta }_{1}\mathrm{Wage}+\sum_{1}^{j}\beta_{j}{\mathrm{Control}}_{j}+\varepsilon$$

Among them, MentalHealth represents the dependent variable (mental health index); Income represents the explanatory variable (the natural logarithm of the individual’s annual income); Wage represents the explanatory variable (hourly wages); Control is a series of other explanatory variables, and the random error term is determined by ε express.

We used the mediating effect model [55] to analyze whether income affects individual mental health through salary satisfaction and interpersonal relationship. The specific formula is as follows:(4)$${\mathrm{MentalHealth}=\beta }_{0}{+\beta }_{1}X+\sum_{1}^{j}\beta_{j}{\mathrm{Control}}_{j}{+\varepsilon }_{1}$$
(5)$$M=\gamma_{0}+\gamma_{1}X+\sum_{1}^{j}\lambda_{j}{\mathrm{Control}}_{j}{+\varepsilon }_{2}$$
(6)$$\mathrm{MentalHealth}=\delta_{0}+\delta_{1}X+\delta_{1}^{\prime }M+\sum_{1}^{j}\delta_{j}{\mathrm{Control}}_{j}{+\varepsilon }_{3}$$

In Formula (4), X is the independent variable (person’s annual income or hourly wage); in Formula (5), M is the mediator variable (economic status and happiness). Equation (4) is used to estimate the total effect of income on individual mental health; Equation (5) is used to estimate the allocation effect of income on intermediary factors; Equation (6) is used to estimate the direct effect and mediation of income on individual mental health effect.

### 3.5. Empirical Results

When investigating the relationship between income and mental health, a person’s smoking and drinking behaviors are usually highly correlated, and the high correlation between variables raises concerns about multicollinearity, which may lead to larger biases. We use the variable inflation factor (VIF) to check for multicollinearity in our model. Table 2 reports the VIF of each variable. In each case, the VIF was less than the empirical value of 10, indicating that multicollinearity was not a major problem.

Table 3 reports the OLS regression results regarding the effect of income on individual mental health. Columns (1) and (3) include only the individual’s annual income and hourly wages, respectively. The regression coefficient of income and wage were 0.529 and 0.511, *p* < 0.01. The effects of columns (1) and (3) showed that increase in income had a significant positive effect on individual mental health. When controlling for a set of covariates in columns (2) and (4), the regression coefficient of income and wage were 0.308 and 0.318, *p* < 0.01. The results showed that income is a significant predictor of individual mental health, showing a positive correlation.

The coefficients of control variables were statistically significant in columns (2) and (4). Specifically, the coefficients of gender and marital status were positive and statistically significant. The results showed that men’s mental health was better than women’s, and married adults had higher mental health than unmarried adults. In addition, the results showed that people with non-agricultural household registration had better psychological status than those with agricultural household registration; people engaged in non-agricultural work had better psychological status than those engaged in agricultural work. Smoking, playing online games, and medical expenses had significant negative effects on mental health. The coefficients of education are positive and statistically significant, indicating that education had a positive effect on individual mental health.

Additionally, we examine the non-linear effects of income on mental health. From the results in columns (2) and (4) of Table 4, it can be seen that the square term coefficient of annual income and hourly wage is positive, and the cubic term coefficient is negative, and both pass the 1% statistical level test. It shows that there is a nonlinear relationship between annual income, hourly wage, and mental health level. Specifically, the relationship between annual income and mental health level shows an “*N*”-shaped curve relationship. By calculating the point where the first derivative of the estimation equation is zero, two values (8.673 and 11.561) are obtained to judge the monotonic change of income affecting mental health. In detail, when the annual income is low, as the income increases, the mental health level decreases; When the income exceeds 8.673, the increase of income has a promoting effect on the level of mental health; As income continued to increase, when the income exceeded 11.561 in the year, the increase in income began to have an inhibitory effect on the level of mental health. Similarly, the relationship between hourly wages and mental health level also shows an “*N*”-shaped curve relationship, with inflection points of 0.361 and 4.123. The results in Table 3 and Table 4 verify Hypothesis 1.

### 3.6. Robustness Check

In the robustness test, we apply the logit model and the probit model to estimate the effect of income on individual mental health. To do this, we substitute a dummy variable for the individual’s mental health. This false value takes 1 if the mental health value is greater than the mean of mental health index, and 0 otherwise. Table 5 shows the estimated results, the personal income and wage coefficients are both positive and statistically significant. That is, increased income has a positive effect on mental health. All in all, the results in Table 5 are basically consistent with those in Table 3.

### 3.7. Heterogeneity

To better understand the relationship between income and mental health, we examine the heterogeneity of effects by dividing the sample into different education state and type of registered residence.

First, the research samples were divided into a Less-educated group and Well-educated group (if the highest level of education is high school or above, the individual is Well-educated, otherwise they are Less-educated). Table 6 presents the results of the heterogeneous effect of income on mental health at different education state. The results of the study show that people with lower education levels show a stronger response to changes in income, regardless of whether the study is based on individual annual income or hourly wage income.

Secondly, the research samples are divided into rural residence registration and non-rural residence registration. Table 7 presents the results of the heterogeneous effect of income on mental health at different type of residence registration. The results show that people with agricultural residence registration are more sensitive to income changes than people with non-rural residence registration, regardless of individual annual income or hourly wage as the research object. The results in Table 6 and Table 7 validate Hypothesis 2.

### 3.8. Mechanisms

In order to explore the mechanism by which income affects mental health, this section uses the mediating effect model to analyze whether income affects individual mental health through the relationship between economic status and happiness.

First, we estimated the effect of individual annual earnings, hourly wage on economic status, and economic status on mental health by OLS. Given that economic status is recorded on an ordinal scale, we also employed an ordinal probit model to investigate the effect of personal income on mental health. Columns (1)–(4) of Table 8 report the effect of individual’s annual income and hourly wage on economic status, and columns (5)–(6) report the effect of economic status on mental health. As can be seen from Table 8, positive and statistically significant coefficients are shown. This suggests that income has an effect on individual’s mental health through the mechanism of economic status.

Furthermore, we estimated the effect of personal annual income, hourly wage on happiness, and the effect of happiness on mental health by OLS. Given that happiness is recorded on an ordinal scale, we also employ an ordinal probit model to investigate the effect of income on happiness. Columns (1)–(4) of Table 9 report the effect of individual’s annual income and hourly wage on happiness, and columns (5)–(6) report the effect of happiness on mental health. As can be seen from Table 8, positive coefficients are reported. Columns (1), (2), and (4) are all statistically significant. This suggests that income has an effect on mental health through its impact on happiness. The results in Table 7 and Table 8 validate Hypothesis 3.

## 4. Discussion

Since its outbreak in 2019, coronavirus (COVID-19) has rapidly become a global health threat and has attracted a lot of attention and research. Among them, many studies are related to the content we study in this article, but their studies tend to focus on specific populations and regions. For example, Brasso et al. [56] conducted a meta-analysis of the existing literature and found that the COVID-19 pandemic has produced a significant impact on the mental health of young people. In addition, Rapisarda et al. [57] analyzed mental health symptoms of mental health workers in Lombardy and Quebec during COVID-19. However, the above studies are based on descriptive studies and lack empirical support. Our study employs ordinary least squares to investigate the relationship between income and individual mental health nationwide and to provide empirical evidence for the effect of income on individual mental health. The results of this paper were consistent with previous studies [7]. However, most studies used non-probability samples. Unlike the study by Pais-Ribeiro et al. [58], our study is derived from large-scale micro-population survey data (CFPS). Furthermore, our study not only investigated the effect of income on individual’s mental health, but also sought to identify the mechanisms between income and mental health.

Despite growing evidence that income may have a positive effect on individual mental health, little has been said to discuss the heterogeneity effect on individual mental health. Our research shows that income affects individuals’ mental health differently, depending on the individual’s education level and type of residence registration. Our research found that people with less education showed a stronger response to changes in income, regardless of whether they were based on annual personal income or hourly wages. In addition, people with agricultural residence registration are more sensitive to income changes than people with non-rural residence registration. Last but not least, our findings provide new evidence for the mechanism between income and individual mental health.

However, this paper has limitations in some aspects. First, the sample data is cross-sectional, limiting our ability to draw causal conclusions. We can only estimate the short-term effects of income on an individual’s mental health, not taking into account the long-term effects. Second, although we try our best to include factors that may affect individuals’ mental health, the model cannot include some hard-to-measure external factors that affect individuals’ mental health. Third, aside from data limitations, we measure mental health in a very general way. An interesting avenue of future research can be investigating the effects of income on individual mental health before the COVID-19 pandemic and make a comparison.

Several policy implications can be derived from this analysis. The empirical results show that income has a significant positive effect on an individual’s mental health. Therefore, governments and society as a whole may need to provide assistance to low-income people during the COVID-19 pandemic. This help should be directed not only to financial subsidies, but also to effective mental health care. The central and local government should input more resources for people with fragile mental health. Nonprofit public organizations such as China Charity Federation should pay particular attention to those who are unemployed and encountering life difficulties due to the COVID-19 pandemic. Social workers are encouraged to conduct more home visits and provide more mental health services. Considering that the income of a less-educated person and a person with rural residence registration is more easily affected by the COVID-19 pandemic, the government should provide as many as possible jobs for them. All types of enterprises, including small enterprises, should try not to lay off staff and survive the temporary difficulties with employees by cutting down unnecessary expenditure. With the efforts of all parties, the goals that everyone’s income will not decline and mental health will not deteriorate would be fulfilled during the COVID-19 pandemic.

## 5. Conclusions

Given the importance of mental health in daily life, there has been an increasing amount of research on this topic during the COVID-19 pandemic. This paper uses 8049 items of data from the 2020 CFPS survey to investigate the causal relationship between income and mental health. One of the most important findings to emerge from the paper is that income had a positive effect on mental health during the pandemic.

The effect of heterogeneity was further analyzed by segmenting the sample according to education level and type of residence registration. On the one hand, the findings show that changes in income have a slightly stronger effect on people with less education than those with more education. On the other hand, those who are not living in rural households respond less strongly to income changes than those who are living in rural households. Our study also investigates the mechanisms by which income affects mental health, showing that income affects mental health through its effects on economic status and happiness.

## Figures and Tables

**Table 1 ijerph-19-08944-t001:** Descriptive statistics of the key variables.

Variable	Definition	Mean	SD	Min	Max
Mental health	Individual’s mental health	−0.028	4.098	−21.165	4.537
Income	Income for the past year (in log)	10.358	0.878	7.313	12.255
Wage	Per hour wage (in log)	2.541	0.961	−1.138	6.070
Sex	1 for male, 0 for female	0.598	0.490	0	1
Age	Individual’s age	39.402	12.110	16	83
Marital	Dummy variable equals 1 if the individual is married, and otherwise 0	0.774	0.418	0	1
registered residence	Dummy variable equals 1 if the residence registration status is agricultural household, and otherwise 0	0.772	0.419	0	1
Smoke	Dummy variable equals 1 if the individual has smoked in the past month, and otherwise 0	0.344	0.475	0	1
Drink	Dummy variable equals 1 if the individual drank alcohol more than 3 times per week, and otherwise 0	0.153	0.360	0	1
Short video	Dummy variable equals 1 if the individual has watched a short video in the past week, and otherwise 0	0.669	0.470	0	1
Online game	Dummy variable equals 1 if the individual has played an online game in the past week, and otherwise 0	0.217	0.412	0	1
Medical expenses	Individual’s medical expenses (in log)	3.115	3.324	0	9.798
Education	Dummy variable equals 1 if the individual has a high school degree or above, and otherwise 0	0.433	0.496	0	1
Identity of worker	Dummy variable equals 1 if the individual is formally employed by the government and state-owned enterprises, and otherwise 0	0.099	0.299	0	1
Type of work	Dummy variable equals 1 if the individual’s is involved in agricultural production, and otherwise 0	0.137	0.343	0	1
Economic status	Rank of revenue in the local area	2.853	0.949	1	5
Happiness	Individual’s happiness	7.400	2.018	0	10

**Table 2 ijerph-19-08944-t002:** The variance inflation factor of each variable.

Variable	VIF	VIF
Income	1.26	
Wage		1.19
Sex	1.69	1.65
Age	1.76	1.74
Marital	1.26	1.25
Registered residence	1.29	1.29
Smoke	1.52	1.52
Drink	1.14	1.14
Short video	1.22	1.22
Online game	1.23	1.23
Medical expenses	1.01	1.01
Education	1.45	1.47
Indentity of worker	1.20	1.20
Type of work	1.16	1.12
Mean VIF	1.32	1.31

Note: VIF represents variable inflation factor.

**Table 3 ijerph-19-08944-t003:** OLS results of the effects of income on mental health.

	(1)	(2)	(3)	(4)
Income	0.529 ***	0.308 ***		
	(10.23)	(5.43)		
Wage			0.511 ***	0.318 ***
			(10.84)	(6.32)
Sex		0.754 ***		0.778 ***
		(6.43)		(6.70)
Age		0.00766		0.00610
		(1.58)		(1.27)
Marital		0.956 ***		0.956 ***
		(8.05)		(8.07)
Registered residence		−0.197		−0.173
		(−1.64)		(−1.44)
Smoke		−0.554 ***		−0.543 ***
		(−4.81)		(−4.72)
Drink		0.0459		0.0452
		(0.35)		(0.34)
Short video		−0.0991		−0.0951
		(−0.95)		(−0.92)
Online game		0.277 *		0.259 *
		(2.32)		(2.17)
Medical expenses		−0.199 ***		−0.197 ***
		(−14.81)		(−14.72)
Education		0.671 ***		0.631 ***
		(6.22)		(5.82)
Identify of worker		−0.413 *		−0.436 **
		(−2.53)		(−2.68)
Type of work		−0.388 **		−0.468 ***
		(−2.80)		(−3.42)
_cons	−5.510 ***	−3.950 ***	−1.327 ***	−1.512 ***
	(−10.25)	(−6.12)	(−10.36)	(−5.11)
*N*	8049	8049	8049	8049
*R* ^2^	0.013	0.060	0.014	0.062
adj. *R*^2^	0.013	0.059	0.014	0.060

Note: t statistics in parentheses. * *p* < 0.05, ** *p* < 0.01, *** *p* < 0.001.

**Table 4 ijerph-19-08944-t004:** Non-linear analysis results.

	(1)	(2)	(3)	(4)
Income	−0.686	−25.660 **		
	(−0.84)	(−2.79)		
Income^2	0.050	2.589 **		
	(1.21)	(2.78)		
Income^3		−0.085 **		
		(−2.73)		
Wage			0.382 *	−0.198
			(2.50)	(−0.71)
Wage^2			−0.014	0.298 *
			(−0.45)	(2.29)
Wage^3				−0.044 *
				(−2.47)
Control variable	YES	YES	YES	YES
_cons	0.950	82.050 **	−1.574 ***	−1.381 ***
	(0.23)	(2.73)	(−4.82)	(−4.11)
*N*	8049	8049	8049	8049
*R* ^2^	0.061	0.061	0.062	0.062
adj. *R*^2^	0.059	0.060	0.060	0.061

Note: t statistics in parentheses. * *p* < 0.05, ** *p* < 0.01, *** *p* < 0.001. ^ is a power sign.

**Table 5 ijerph-19-08944-t005:** Robustness test results.

Variable	OLS	Logit	Probit	OLS	Logit	Probit
Income	0.0304 ***	0.128 ***	0.0795 ***			
	(4.38)	(4.36)	(4.38)			
Wage				0.0337 ***	0.142 ***	0.142 ***
				(5.48)	(5.44)	(5.44)
Control variable	YES	YES	YES	YES	YES	YES
_cons	0.121	−1.598 ***	−0.993 ***	0.356 ***	−0.612 ***	−0.612 ***
	(1.53)	(−4.79)	(−4.80)	(9.84)	(−3.98)	(−3.98)
*N*	8049	8049	8049	8049	8049	8049
*R* ^2^	0.043			0.044		
adj. *R*^2^	0.041			0.042		

Note: t statistics in parentheses. *** *p* < 0.001.

**Table 6 ijerph-19-08944-t006:** Heterogeneous effects of different education levels on income.

Variable	Less-Educated	Well-Educated	Less-Educated	Well-Educated
Income	0.382 ***	0.178 *		
	(4.95)	(2.09)		
Wage			0.365 ***	0.213 **
			(5.33)	(2.76)
Control variable	YES	YES	YES	YES
_cons	−5.001 ***	−1.739	−1.900 ***	−0.449
	(−5.70)	(−1.79)	(−4.57)	(−1.11)
*N*	4563	3486	4563	3486
*R* ^2^	0.073	0.035	0.074	0.036
adj. *R*^2^	0.071	0.032	0.072	0.033

Note: t statistics in parentheses. * *p* < 0.05, ** *p* < 0.01, *** *p* < 0.001.

**Table 7 ijerph-19-08944-t007:** Heterogeneous effects of different exercise frequencies on income.

Variable	Rural	Non-Rural	Rural	Non-Rural
Income	0.321 ***	0.280 *		
	(5.01)	(2.20)		
Wage			0.328 ***	0.275 *
			(5.64)	(2.50)
Control variable	YES	YES	YES	YES
_cons	−4.288 ***	−3.555 *	−1.725 ***	−1.294 *
	(−6.08)	(−2.48)	(−5.84)	(−2.28)
*N*	6215	1834	6215	1834
*R* ^2^	0.060	0.050	0.062	0.051
adj. *R*^2^	0.059	0.044	0.060	0.045

Note: t statistics in parentheses. * *p* < 0.05, *** *p* < 0.001.

**Table 8 ijerph-19-08944-t008:** With economic status as a mediating variable.

Economic Status					Mental Health	
	(1)	(2)	(3)	(4)	(5)	(6)
Variable	OLS	OLS	Ordinal Probit	Ordinal Probit	OLS	OLS
Income	0.139 ***		0.163 ***		0.255 ***	
	(10.31)		(10.49)		(4.45)	
Wage		0.110 ***		0.129 ***		0.273 ***
		(9.22)		(9.37)		(5.39)
Economic status					0.494 ***	0.492 ***
					(10.44)	(10.42)
Control variable	YES	YES	YES	YES	YES	YES
_cons	1.209 ***	2.385 ***			−4.732 ***	−2.731 ***
	(7.90)	(34.12)			(−7.28)	(−8.64)
*R* ^2^	0.035	0.033			0.074	0.075

Note: t statistics in parentheses. *** *p* < 0.001.

**Table 9 ijerph-19-08944-t009:** With happiness as a mediating variable.

Happiness					Mental Health	
	(1)	(2)	(3)	(4)	(5)	(6)
ariable	OLS	OLS	Ordinal Probit	Ordinal Probit	OLS	OLS
Income	0.0488		0.0167		0.280 ***	
	(1.73)		(1.13)		(5.15)	
Wage		0.0626 *		0.0240		0.281 ***
		(2.50)		(1.83)		(5.84)
Happiness					0.585 ***	0.584 ***
					(27.27)	(27.21)
Control variable	YES	YES	YES	YES	YES	YES
_cons	6.510 ***	6.867 ***			−7.759 ***	−5.521 ***
	(20.27)	(46.64)			(−12.25)	(−17.30)
*R* ^2^	0.041	0.042			0.140	0.141

Note: t statistics in parentheses. * *p* < 0.05, *** *p* < 0.001.

## Data Availability

The CFPS data can be accessed through its official website (http://www.isss.pku.edu.cn/cfps/download/index#/fileTreeList (accessed on 4 May 2022).

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
