# Peer review of "Relationship between Income and Mental Health during the COVID-19 Pandemic in China"

_ijerph, 2022, doi:10.3390/ijerph19158944_

Round 1

Reviewer 1 Report

The manuscript titled: “Impact of Income on Mental Health during the COVID-19 Pan- 2 demic: Based on the 2020 China Family Panel Survey addresses an important mental health issue during the COVID-19 pandemic”. Research, in fact, studies the overall impact of personal income on the mental health of individuals on mental health trying to show that income can affect mental health through its impact on salary satisfaction and interpersonal relationships.

The manuscript is well-written and of interest to the scientific community, as by attempting to demonstrate that income can influence mental health through its impact on salary satisfaction and interpersonal relationships, it provides information on the mechanisms on how income influences mental health.

Research confirms the influence of social determinants on health, therefore the following article is suggested for inclusion in the bibliography:

1.    Ferrara M, Langiano E, Falese L, De Marco A, De Vito E. Quality of Life and Psychosocial Impacts of the Different Restrictive Measures during One Year into the COVID-19 Pandemic on Patients with Cancer in Italy: An Ecological Study. Int J Environ Res Public Health. 2021 Jul 4;18(13):7161. doi: 10.3390/ijerph18137161. PMID: 34281098; PMCID: PMC8297179.

Author Response

Thank you for your valuable comment. As you said, the influence of social determinants on health is very important. We add the literature in my paper. The revisions were marked in the text.

Ferrara et al. [6] reported that the COVID-19 disease and the restrictive government measures are harming the mental health of individuals around the world, causing fear, frustration, anger and a long list of complex negative emotions.

Reviewer 2 Report

1.        As an academic paper, there should be a separate section of literature review, which is positive to address the significance and cutting-edge of this study.

2.        The data used in this study is CFPS 2020. However, this data mainly survey the household status before COVID-19, so the research background may not be appropriate. The authors are encouraged to revise the research background and the title.

3.        One concern of this paper is that the cross-sectional data and the methodologies used in the paper do not imply the conclusions made based on causality. The authors are recommended to include the prior waves of CFPS data to produce more robust estimates.

4.        The hypotheses need to be tested in the baseline estimations, and a heterogeneity test is used to verify the estimation results. Nevertheless, the authors used heterogeneity analysis to test hypothesis 2. The logic is confusing.

5.        To address the mediating effect, in Column (9) of Tables 8 and 9, the variable of income should also be incorporated. Meanwhile, the estimation method should be consistent with prior ones.

6.        The authors are encouraged to address the implications and limitations of this study.

Author Response

The revisions above we have made are according to the your comments. We appreciate their insights and hope that the corrections meet your expectations. Once again, thank you very much for  the valuable comments and helpful suggestions.

Reviewer 3 Report

I recommend to be more specific as far as companies’ possible strategies to help employees to feel better. You clearly proved that income has an impact on depression and (un)happiness. Also stated that physical activities can reduce these negative feelings. BUT specifically, what can non public organizations do to motivate employees and help them feel good? Make available a gym? Pay for it? Health insurance offer? Bonus for those that look for help? Because, in fact, your article is interesting and important, not only in a clinical perspective, but also in an employment/management perspective. So, please try to state specific recommendations on how employers can help employees. The same for public institutions. My second recommendation is to explain/prove precisely that the sample is representative of the country’s population.

In general, I find your article very interesting and actual.

Author Response

Thank you for your valuable comment. Though non public organizations are not well developed in China, it is important to play a vital role. Also, employers are important from the employment/management perspective. Following your suggestion, we make the revision as follow.

Therefore, governments and society as a whole may need to provide assistance to low-income people during the COVID-19 pandemic. This help should be directed not only to financial subsidies, but also to effective mental health care. The central and local government should input more resources for people with fragile mental health. Nonprofit public organizations such as China Charity Federation should pay particular attention to those who are unemployment and encountering the life difficulties due to the COVID-19 pandemic. Social workers are encouraged to conduct more home visits and provide more mental health services. Considering the income of less educated person and person with rural residence registration is more easily affected by the COVID-19 pandemic, government should provide as many as possible jobs for them. All types of enterprises, including small enterprises should try not to lay off staff and survive the temporary difficulties with employees by cutting down unnecessary expenditure. With the efforts of all parties, the goals that everyone's income will not decline and mental health will not deteriorate would be fulfilled during the COVID-19 pandemic.

Round 2

Reviewer 3 Report

After carefully reading this "improved" version of the article I became even more convinced of the importance and pertinence of the topic.

Further studies should include Managers' perspectives on this subject as well... but this comment is for another research, not for the present one.